# Recovery of Diamond and Cobalt Powder from Polycrystalline Drawing Die Blanks via Ultrasound-Assisted Leaching Process—Part 1: Process Design and Efficiencies

**Ferdinand Kießling** [1], **Srecko Stopic** [2,*], **Sebahattin Gürmen** [3] **and Bernd Friedrich** [2] 

1    Redies Deutschland GmbH & Co.KG, Metzgerstr. 1, 52070 Aachen, Germany; fer87den@gmail.com
2    IME Process Metallurgy and Metal Recycling, RWTH Aachen University, Intzestrasse 3, 52056 Aachen, Germany; bfriedrich@ime-aachen.de
3    Metallurgical and Materials Engineering Department, Istanbul Technical University, Ayazaga Campus, Istanbul 34469, Turkey; gurmen@itu.edu.tr
*    Correspondence: sstopic@metallurgie.rwth-aachen.de; Tel.: +49-24-1809-5860

**Abstract:** The treatment of industrial polycrystalline diamond (PCD) blanks in aqua regia at atmospheric pressure between 333 K and 353 K was performed via the ultrasound-assisted leaching process to investigate whether the influence of ultrasound is beneficial. Cobalt content in the solution and in the blanks was monitored as well as the effects of leaching temperature, solid-to-liquid ratio, and PCD blank size. The use of intermittent and permanent ultrasound helped reduce the leaching time and thus energy consumption by up to 50%. In all trials with ultrasound, higher temperature only has a slight effect. Solid-to-liquid ratio does not have a positive or negative impact. A new process design was tested using an innovative experimental setup for ultrasound-assisted leaching aiming at maximum cobalt and diamond recovery from PCD and final reuse of fine PCD for cutting and polishing other hard materials in different important industrial applications.

**Keywords:** polycrystalline diamond; leaching; cobalt; ultrasound

## 1. Introduction

Cobalt is a ferromagnetic transition metal which is located between iron and nickel in the periodic table of elements and mostly available in lateritic ores [1–3]. Cobalt is used as a solvent catalyst in the production of polycrystalline diamond (PCD) that would otherwise take even more pressure and a higher temperature to achieve. [4]. The wide use of cobalt relative to other metals is associated with the high solubility of carbon in its melt during thermobaric treatment [5]. Unfortunately, at temperatures above 800 °C, which are developed during operation of a polycrystalline diamond tool, the cobalt promotes the formation of microcracks and results in significant heat-resistance reduction and subsequent reduction in the abrasion resistance of the polycrystalline diamond [6]. Thus, the reduction of the cobalt content in the sintered polycrystalline diamond (PCD) greatly improves the performance of a superhard composite. Finally, the removal of cobalt from the used PCD is the most important request in the industry of wire-drawing tools made from diamond materials and tungsten carbide.

Therefore, recycling is a chosen strategy for cobalt recovery in contrast to traditional primary metallurgy [7]. Diamond, to this date, is the hardest material that is put to use on commercial and industrial scales. It appears as the working edges of cutting tools or the grains in the hardest abrasives, among other uses. Due to advancements in the field of high strength steels and super alloys, the demand for hard cutting and forming tools will be increased in the future [8]. Industrially used diamonds can be

found in the form of a naturally grown and mined crystals or as a man-made products with mono- or poly-crystalline microstructures. Die blanks are usually made at high temperatures and high pressure processes which require cobalt (Co) as the solvent catalyst [9]. Cobalt is incorporated in the final product resulting in a multiphase compound (PCD). This multiphase characteristic renders the PCD vulnerable to thermal stress since cobalt and diamond have different thermal expansion coefficients. The only way to make these PCDs stable for industrial applications at high temperatures, such as hot forming of metals, is to remove the inclusions from the cavities in the framework of diamond grains. Because of its strong oxidizing properties, aqua regia, or more precisely the forming nitrosyl chloride (NOCl), was used for the leaching of cobalt. This research also acknowledges the importance of using cobalt responsibly, possibly in a closed-loop recycling, not only because of price and toxicity but rather socioeconomic problems and difficulties linked to the conflict mineral columbite–tantalite, often referred to as 'coltan' [10,11].

Since the middle of the twentieth century, there have also been investigations into whether ultrasound can increase the chemical turnover in leaching processes. Many researchers have found that the reaction rate as well as the overall leaching efficiency can be increased significantly [12–14]. This positive effect is attributed to cavitation and specifically the Kelvin impulse [15–17]. Ultrasound is capable of creating such dynamic pressure changes in liquids that they will pass the phase boundary to their gaseous phase. When located near a solid surface, these cavities collapse and create an energetic jet pointed at the surface, because there is less instreaming liquid from the direction of the surface and overall impulse has to be conserved. These microjets have the ability to pierce through diffusion layers, effectively constantly renewing them. The positive influence of ultrasound was confirmed using ultrasound for the synthesis of nanosized particles by ultrasonic spray pyrolysis (USP). Due to its easy feasibility, flexibility and cost-efficiency, the USP method is an important alternative to the chemical vapor deposition (CVD) and other synthesis methods. Cobalt nanoparticles were successfully prepared from cobalt nitrate solution formed after an acidic treatment of the cemented tungsten carbide using the ultrasonic spray pyrolysis method [18,19]. An increase of ultrasound from 0.8 to 2.5 MHz decreases an aerosol diameter of cobalt nitrate to 2.2 μm that leads to the formation of submicron cobalt particles after drying and precipitation above 500 °C in a furnace using a hydrogen reduction atmosphere.

Shortening the process time for the cobalt removal from PCD in the presence of ultrasound is the main motivation for this study. In solid–liquid reactions, there is always an obstacle in the form of a diffusion layer where the mass transfer is inhibited because adsorption reaction and desorption take additional time and energy; even more so if a phase transition is involved. What's more, the component's concentrations can be quite different from the bulk solution, hindering further reaction. One way to mitigate this effect is to mechanically agitate the solution and decrease the thickness of this layer. For this reason, a stirrer was used to create sufficient turbulence in the reaction vessel.

An increase in temperature has an effect on most chemical reactions as it measures a substance's inner energy that is potentially available for reactions, changes the substance's activity or simply aids the mass transfer processes. In the leaching process, it has been found that increased temperature can accelerate the reaction speed [20–22]. The dissolution process with an acid needs a high activation energy to begin this solid–liquid reaction. Since this study is carried out at ambient pressure with the leaching solution described, care has to be taken because increasing the temperature also increases vapor pressures of the liquids involved [23].

Finally, the main aim of this work is optimizing leaching of cobalt from polycrystalline diamond blanks with grain size 5 μm in the presence of ultrasound. The influence of temperature and ultrasound on the leaching of cobalt will be studied in one ultrasound leaching assisted process. This study will take a look at the effects of ultrasound on the leaching efficiency but also at the penetration depth into the PCD, proposing a new experimental setup. Being capable of making any statements about the metallic constituents of PCD without breaking the blanks is an advantage and the reason why this method has been implemented. A challenge of this work is to maximize efficiency of cobalt leaching and diamond recovery in a shorter time than traditional hydrometallurgical methods.

## 2. Experimental

### 2.1. Material

As shown in Figure 1, this study is centered on the polycrystalline diamond (PCD) blanks made by Redies GmbH & Co. KG, Aachen, Germany, a manufacturer of wire drawing dies.

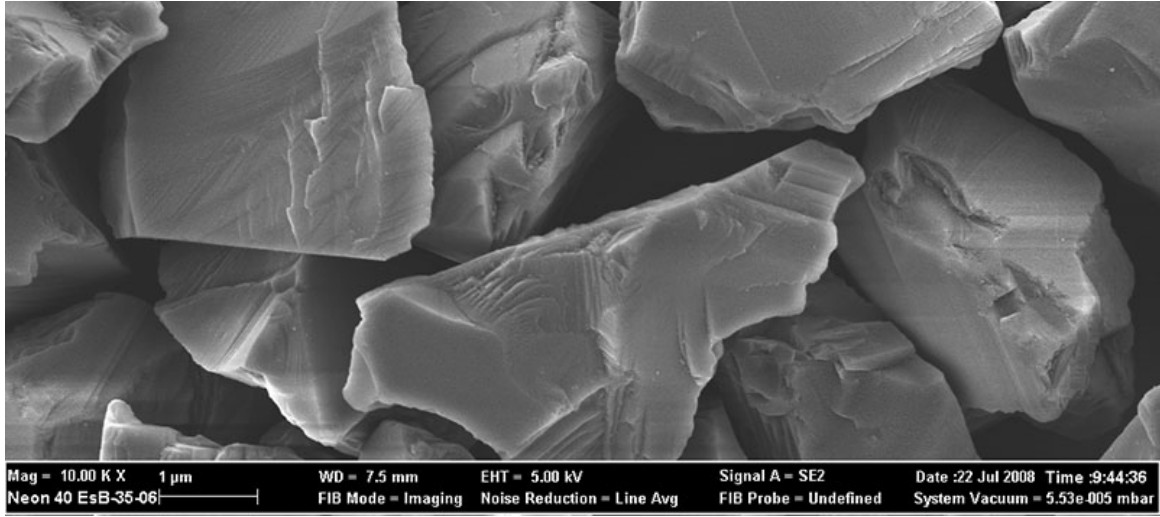

**Figure 1.** Scanning electron microscopy (SEM) image of raw diamond powder 5 μm class before the high temperature and high pressure (HTHP) process.

The SEM (Scanning electron microscopy) and EDS (Energive Dispersive Spectroscopy) analysis of the PCD surface after being polished is shown in Figure 2 and Table 1.

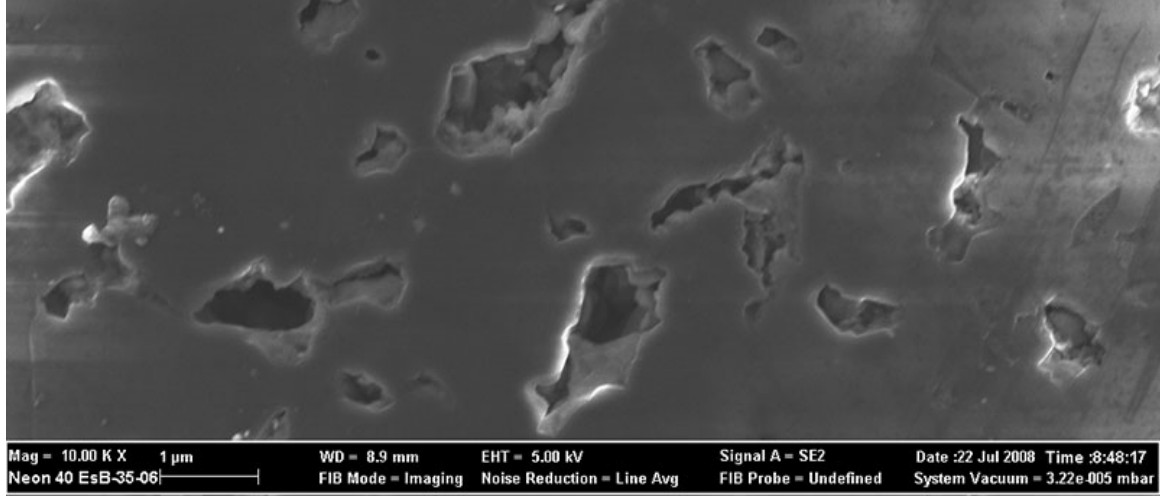

**Figure 2.** SEM image of ground and polished PCD surface, 5 μm class (dark grey areas are the bridged diamond grains with cavities where also traces of cobalt show up in lighter shades).

In contrast to the cobalt content of 0.1 wt.% in nickel lateritic ore, its average content in PCD is about 5–20 wt.%, which makes it a very promising material for the recycling of cobalt. The maximal value of cobalt in analyzed sample amounts is 1.67 wt.%, as shown in Table 1.

**Table 1.** Energy dispersive X-ray spectroscopy image of ground and polished PCD surface, also 5 μm class.

| Content (Weight %) | C | O | Fe | Co |
|---|---|---|---|---|
| Spectrum 1 | 86.81 | 12.94 | - | 0.25 |
| Spectrum 2 | 88.31 | 10.42 | 0.21 | 1.05 |
| Spectrum 3 | 89.35 | 10.32 | - | 0.34 |
| Spectrum 4 | 87.41 | 10.78 | 0.14 | 1.67 |
| Max. | 89.35 | 12.94 | 0.21 | 1.67 |
| Min. | 86.81 | 10.32 | 0.00 | 0.25 |

Different types of samples were used in our work as shown in Table 2. The columns "volume", "surface area", and "surface-to-volume ratio" in Table 2 do not contain measured but calculated values. The values for volume should be seen as the apparent outer volume of a porous body, not solid volume. The same applies to surface area. The measured values in the columns "diameter" and 'height' were obtained by taking a sample of thirty PCDs of the same type and averaging the values. The value for weight was obtained by weighing 100 PCDs and dividing the measurement by 100. This approach was chosen since the blanks did not have critical deviations dimension-wise. Assuming a homogeneous density, this average is sufficiently accurate. The same applies to weighing the batches as a whole after treatment. Figure 3 shows the deviations in the weights of individual PCDs. All the weight measurements were taken into a stock plot to depict actual variations and uncertainties. The deviations of surface-to-volume ratio were obtained by relating the largest surface to the smallest volume and vice versa.

**Table 2.** Dimensions of PCD samples with grain size of 5 μm.

| Blank Type | Symbol | Diameter [mm] | Height [mm] | Weight [g] | Volume [mm$^3$] | Surface Area [mm$^2$] | Surface/Volume [mm$^{-1}$] |
|---|---|---|---|---|---|---|---|
| Mant$^®$ MSD-06-005 | DO6 | 2.97 ± 0.01 | 1.10 ± 0.02 | 0.029 | 7.62 ± 0.18 | 24.09 ± 0.28 | 3.16 ± 0.11 |
| Mant$^®$ MSD-14-005 | D14 | 4.05 ± 0.09 | 2.00 ± 0.04 | 0.099 | 25.71 ± 1.40 | 51.14 ± 1.97 | 1.99 (+0.20|−0.18) |
| Mant$^®$ MSD-15-005 | D15 | 5.2 | 2.5 | 0.241 | 53.093 | 83.315 | 1.569 |
| Mant$^®$ MSD-18-005 | D18 | 5.22 ± 0.02 | 3.50 ± 0.02 | 0.299 | 74.90 ± 0.77 | 100.22 ± 0.68 | 1.338 ± 0.023 |

*2.2. Procedure*

The experiments were carried out in two glass reactors simultaneously set up in a fume cabinet, using argon gas (Linde Gas AG, Höllriegelskreuth, Germany) a flow meter, type Rota Yokogawa (Yokogawa Deutschland GmbH, Ratingen, Germany), and a bottle with sodium hydroxide (Merck KGaA, Darmstadt, Germany), as shown at Figure 4. The reactor vessels were three-necked round bottom flasks with a capacity of 500 mL, the necks with standard ground joints 29/32 served as couplings for a stirrer seal, two gas hose couplers, and as access points for sampling, respectively, as shown at Figure 5. For sampling, the gas inlet coupler had to be removed temporarily. The stirrer unit consisted of a motor unit with a drill chuck, type "IKA Eurostar digital" (IKA$^®$-Werke GmbH & Co.KG, Staufen, Germany) IKA$^®$-Werke GmbH & Co. KGIKA$^®$-Werke GmbH & Co. KGIKA and a Polytetrafluorethylen (PTFE) coated impeller including a PTFE stirrer seal, as shown at Figure 4. The standard ground joints on the sides were sealed with a high-viscosity, silicone-based lubricant.

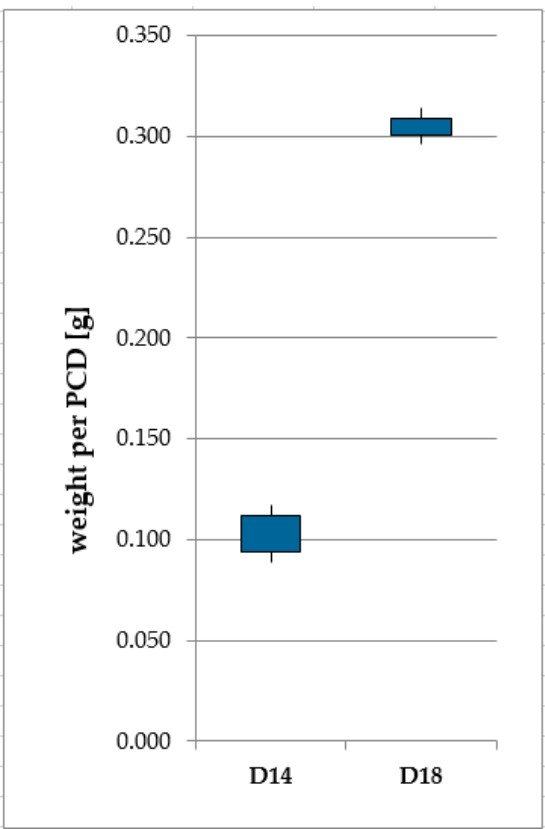

**Figure 3.** Deviations in weight of individual D14 and D18 PCD blanks.

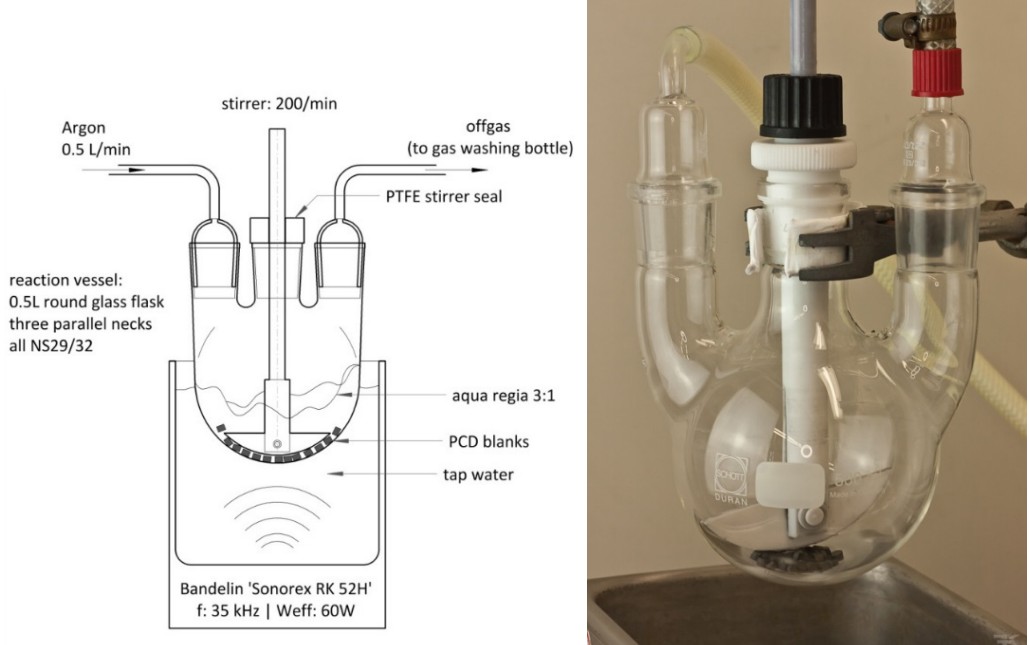

**Figure 4.** The description and picture of the leaching reactor.

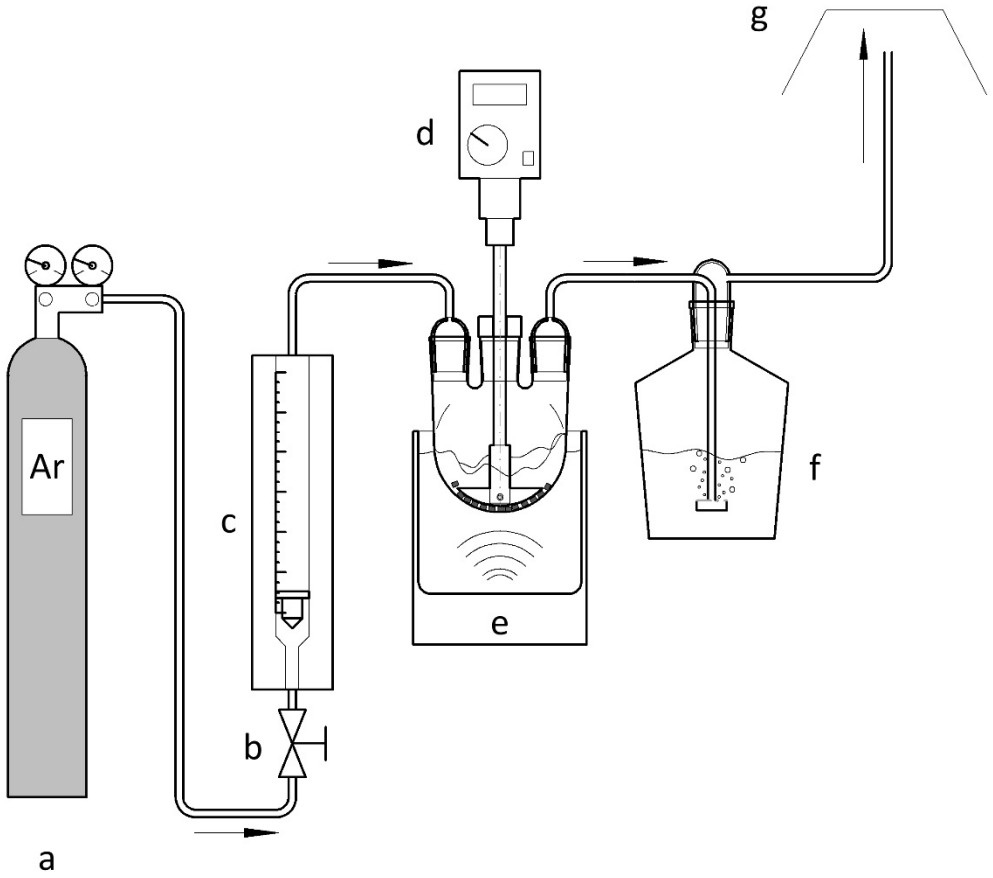

**Figure 5.** Innovative experimental set-up for leaching of PCD samples where: a—argon, b—valve; c—flow meter; d—mixer, e—reactor, f—bottle with dissolved sodium hydroxide, e—exhaust system.

The depth of immersion was chosen so that the surface level of the stirred liquid was as high as the water level in the heated ultrasound bath. The means of hindering evaporation can also be seen in this image. First, styrofoam beads were added to minimize the surface area available for evaporation, and secondly an acrylic lid made of two parts and an improvised cable tie hinge was used. For additional safety, each connection was clamped. For the purpose of temperature value measurement a "Testo 720" digital thermometer (testo SE & Co. KGaA, Lenzkirch, Germany) with "PT100" thermocouple (Temperatur Messelemente TMH, Hettstedt GmbH, Maintal, Germany) was used.

Below the aforementioned setup ultrasonic baths, "Bandelin Sonorex RK 52H" (BANDELIN electronic GmbH &Co. KG, Berlin, Germany) types were placed on lab jacks so they could be lowered for sampling and batch changes. This arrangement made disassembly easier and did not require readjusting the upper structure with every batch change. These ultrasound baths have a nominal frequency of 35 kHz and put out 60 W effectively, while output peaks can occur up to 240 W. The output level was fixed and ultrasound irradiation was altered by using it intermittently. In this case, they were filled with tap water to maximum capacity. The water volume was about 1.1 L due to the volume displaced by the reaction vessels.

As shown in Table 3, the parameters for the leaching experiments were proposed using our previous hydrometallurgical experience and previously performed experiments, reported in the literature [20].

**Table 3.** Parameters for the leaching of cobalt from polycrystalline diamond blanks.

| ID | $T_{Bath}$ [K] | PCD Type | Leaching Time [h/d] in the Presence of Ultrasound | S/L [g/L] |
|---|---|---|---|---|
| W1R1 | 333 | D14 | 0 | 15 |
| W1R2 | 353 | D14 | 0 | 15 |
| W2R1 | 333 | D14 + D18 | 0 | 30 |
| W2R2 | 353 | D14 + D18 | 0 | 30 |
| W3R1 | 333 | D14 + D18 | 0 | 45 |
| W3R2 | 353 | D14 + D18 | 0 | 45 |
| W4R1 | 333 | D14 + D18 | 8 h/d | 15 |
| W4R2 | 353 | D14 + D18 | 8 h/d | 15 |
| W5R1 | 333 | D14 + D18 | 8 h/d | 30 |
| W5R2 | 353 | D14 + D18 | 8 h/d | 30 |
| W6R1 | 333 | D15 + D18 | 8 h/d | 45 |
| W6R2 | 353 | D15 + D18 | 8 h/d | 45 |
| W7R1 | 333 | D14 + D18 | 24 h/d | 15 |
| W7R2 | 353 | D14 + D18 | 24 h/d | 15 |
| W8R1 | 333 | D14 + D18 | 24 h/d | 30 |
| W8R2 | 353 | D14 + D18 | 24 h/d | 30 |
| W9R1 | 333 | D14 + D18 | 24 h/d | 45 |
| W9R2 | 353 | D14 + D18 | 24 h/d | 45 |

The abbreviations in the first column are read as successive week number and R1 and R2 representing reactors 1 and 2, respectively. These codes also served as stems for sample identification. The column header S/L is short for solid-to-liquid ratio in units of grams per liter. Constant parameters were stirring speed and batch time. The duration of each batch was planned to be between 90 and 100 h. Sample names D06, D14, D15 and D18 are abbreviated product names of Mant® MSD-06-005, MSD-14-005, MSD-15-005, and MSD-18-005, all self-supported PCD blanks with diamond grain sizes of 5 μm.

### 2.2.1. Preparation of Samples

The PCD blanks were weighed and measured with the teslameter (Projekt Elektronik GmbH, Berlin, Germany), beforehand to obtain the important '100%' reference value for evaluation. The aqua regia was mixed from three parts fuming hydrochloric acid 37% Emsure ACS/ISO quality (Merck KGaA, Darmstadt, Germany) and one part nitric acid 65% ISO analysis quality PanReac ApplicChem (Chicago, IL, USA). 240 mL of hydrochloric acid and 80 mL nitric acid were prepared in covered beakers for each reactor. Meanwhile, the ultrasound baths were filled with tap water and their heaters were set to 333 K and 353 K, respectively, to ensure that the bath temperature was nominal from the beginning of the experiment. Gas tightness of the apparatus was checked daily.

### 2.2.2. Conduct of Experiments

In weeks 1 through 3, the ultrasound baths were only used as heated water baths. In weeks 4–6 the ultrasound was intended to be switched on for eight hours per day. In addition to refilling water, the time of ultrasound irradiation had to be noted. In the remaining three weeks the ultrasound was switched on permanently. Concerning the forced gas flow, a current of around 0.25 L/min was sufficient to ensure the flow in one direction only. Argon was chosen over nitrogen or pressured air because oxygen and nitrogen might have skewed the equilibria with $NO_2$, $N_xO_x$ or formed combustible mixtures with chlorine gas or hydrogen gas.

### 2.2.3. Sampling

Throughout the experiments, only cobalt content in the solution and changes in magnetic properties of the PCD were sampled each day. First, a few milliliters of the solution were pumped from each reactor using a plastic syringe and PTFE tube and transferred into a small beaker. From there,

1 mL of solution was taken with a pipette and added to a 50 mL round glass flask which was then filled with deionized water up to the 50 mL mark, resulting in a 1 in 50 dilution. This sample solution was transferred again into a 50 mL sample vial, and analyzed by the chemistry department at the IME, RWTH Aachen University (Aachen, Germany) by inductively coupled plasma–optical emission spectrometry (ICP-OES) (SPECTRO ARCOS, SPECTRO Analytical Instruments GmbH, Kleve, Germany). The solid sample was analyzed by X-ray fluorescence (Axios FAST, Malvern Panalytical GmbH, Germany).

The first indication that $CoCl_2$ was formed could be seen when taking samples from the solution. Especially towards Thursday and Friday of any experimental week, the solution taken from the reactor had a dark greenish teal color that changed to pink after a few seconds in the beaker, indicating the typical drying salt color change from the dihydrate ($CoCl_2 \cdot 2H_2O$) to the hexahydrate ($CoCl_2 \cdot 6H_2O$) form of $CoCl_2$ when cooling below approximately 308 K. Liquid samples were analyzed with the method of inductively coupled plasma optical emission spectrometry (ICP-OES).

### 2.2.4. Weighing of Sample

For the purpose of determining the mass balances, PCDs were weighed before and after each batch on the same scale in the laboratory, type LA620P (Sartorius AG, Göttingen, Germany). Before the experiments, the PCDs were weighed as they were delivered. After an experiment they were wet with aqua regia, so they had to be rinsed with distilled water at least twice. In between stages, the PCDs were left in fresh distilled water for about 15 min. After the last rinse they were shaken with a little ethanol to assist in the drying process. Then, after two days at 353 K in the laboratory dryer. They were weighed while still warm. This was to ensure that no humidity would skew the results of weighing. According to Redies GmbH & Co. KG (Aachen, Germany), Aachen, a loss of around 20 weight percent due to extraction of cobalt from PCD is to be expected.

### 2.2.5. Observation of Changes in Magnetic Properties of PCD

Changes in magnetic properties were observed using a teslameter, type FM 205 (Projekt Elektronik GmbH, Berlin, Germany), with a reference neodymium magnet. The apparatus for measuring of magnetic properties is shown in Figure 6.

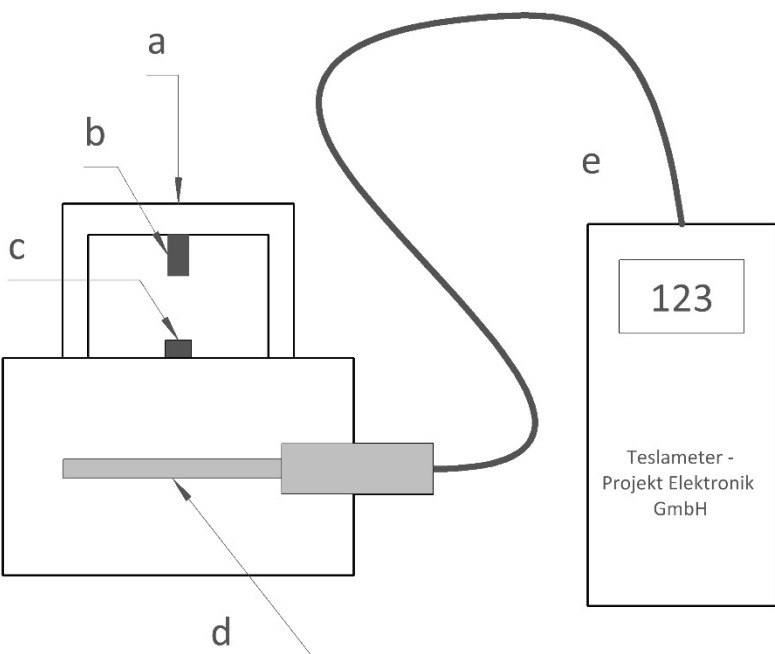

**Figure 6.** Measuring magnetic properties of a PCD. a: Magnet support, b: ∅ 5 mm by 8 mm Nd alloy magnet, c: PCD blank, d: probe, e: handheld teslameter.

It consists of a teslametric probe that is kept at a fixed distance from the reference magnet, such as Nd alloy. The probe will display a value at any time representing the current magnetic situation. To diminish disrupting effects from the surroundings, the probe was offset by the nearby reference magnet. The actual measurement was always a sum of the magnetic surroundings, reference magnet and the subject in between. With only air and plastic between magnet and probe, the reading on the display was 583. A measurement was taken before every run with readings varying above 600. The value displayed for air as a subject is taken as 0% as a reference value. The value for the unleached PCD marks the 100% value for each batch and reactor, respectively. This method allows for a normalization of values and a plot of inferred Co content in the PCD relative to its initial content. In the course of the experiment, this value dropped towards the value of normal air, enabling an estimate of the progress of relative cobalt content in the PCD.

Since all measurements were always compared against the offset value and normalized to the pre-experiment value being 100%, units cancel out. It has to be noted that this offset value was registered before each individual PCD measurement because it changed between 583 and 584—perhaps due to other magnetic influences, or the magnet may have been placed in a way that resulted in a measurement on the threshold between 583 and 584. To ensure consistency, the magnet was not moved until after the experiments. An important caveat is the fact that the relative Co content is inferred, not measured. The change in the magnetic field of the probe consists of more than just the effect due to the presence of cobalt. In this case, analyses have shown that there are oxygen and iron impurities present in the raw PCD. Fe(II), Fe(III), Co(II), and Co(III) oxides have magnetic properties that naturally differ from pure Co. The overall effect of magnetic metals is measured and the Co content is concluded from these values.

## 3. Results and Discussion

### 3.1. Mass Balance

The measured concentration of cobalt in the samples was plotted against the time when the samples were taken to visualize the accumulation in solution, as shown in Figure 7 (from 1 and 9 week).

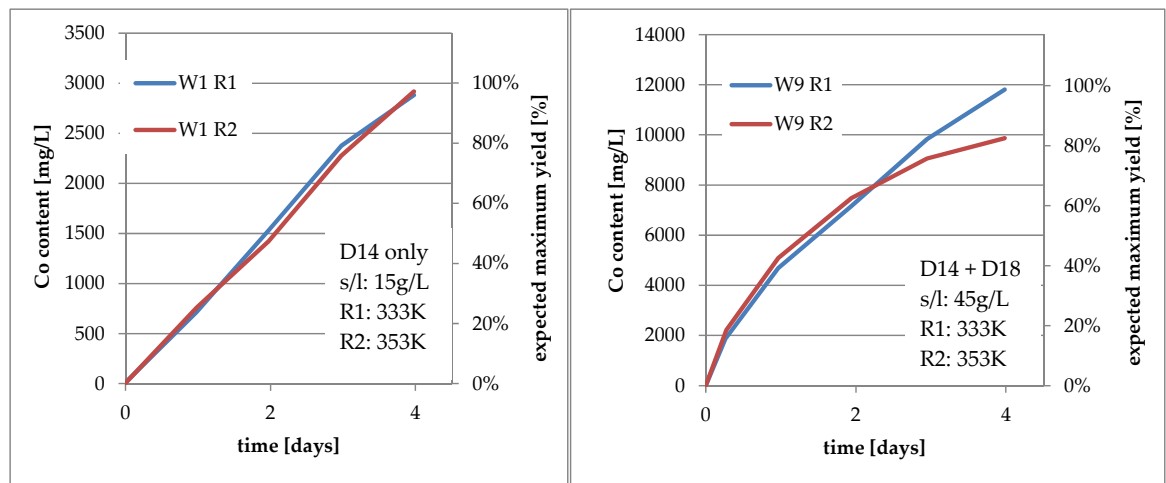

**Figure 7.** Cobalt concentration versus time in weeks 1 and 9.

These plots were chosen as the extreme cases, with the other results very similar and within their range. Without ultrasound and at the low end of solid-to-liquid ratios, the concentration of cobalt developed very similarly regardless of temperature, as shown at Figure 8. However, towards the end of week nine, at a high solid-to-liquid ratio and with full ultrasound, a difference emerged where the Co content in reactor 2 seemed to go into saturation. It appeared from week 4 onwards, so it may be linked to the higher temperature and use of ultrasound.

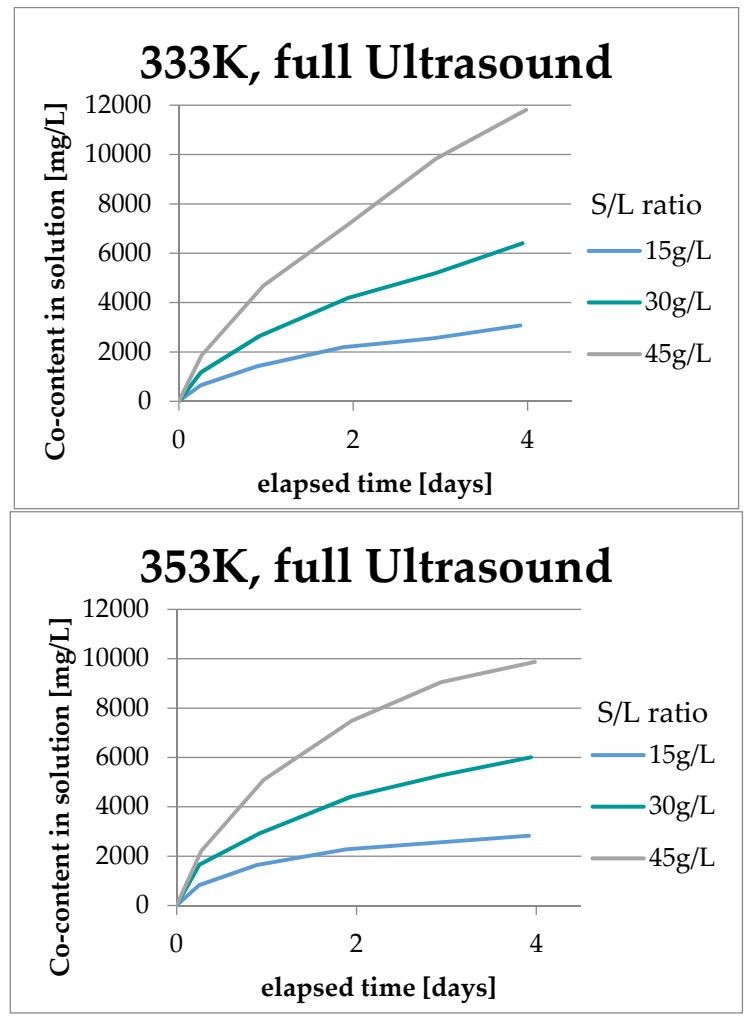

**Figure 8.** Concentration of cobalt in solution in time between 333 K and 353 K with full-time ultrasound.

When comparing the two reactors independently at Figure 8, this discrepancy becomes more obvious. With ultrasound used full-time, the solution at 353 K accumulated several percent less cobalt in total than the cooler reactor. However, the initial increase happened faster. To see the effect of ultrasound itself, the concentration curves from low and high solid-to-liquid ratios were plotted for both reactors, as shown at Figure 9. Interestingly, full-time ultrasound did not seem to achieve the highest cobalt yields. At high solid-to-liquid ratios, it did not even seem to do any better than the experiments without ultrasound.

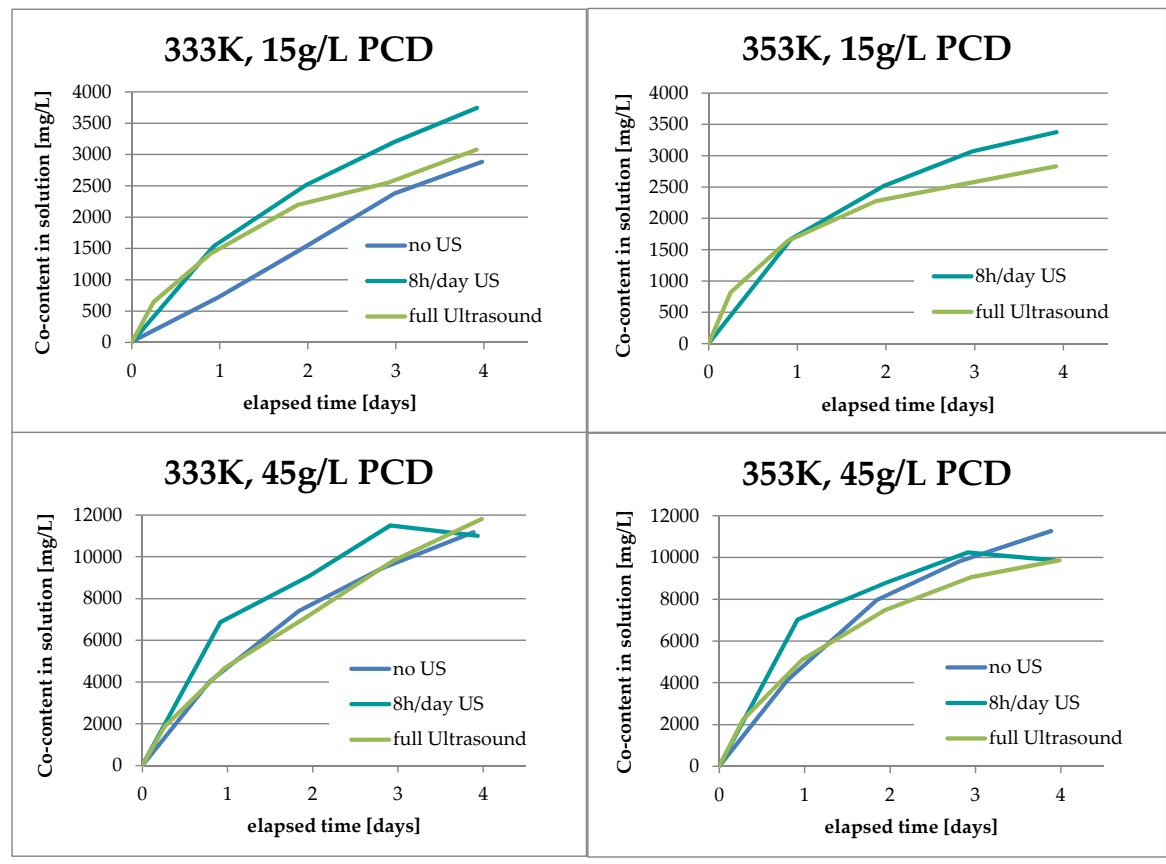

**Figure 9.** Influence of ultrasound on concentration of cobalt in the solution over time.

As stated above, the PCDs were weighed to gain insight into whether the chosen parameters, especially ultrasound, aided the leaching process. These measurements were used to compare the results of all runs, shown individually for D14 and D18 blanks in Figure 10. At first glance, the most obvious fact is that the larger PCDs were not leached to completion within the 90 to 100 h timeframe. The maximum values for lost PCD weight were 21.04% and 13.86% for D14 and D18, respectively. For D14, the most influential parameter appears to be ultrasound. There is a strong influence of bath temperature without it. However, already intermittent ultrasound is enough to drive the leaching efficiency towards the expected value. In the runs with ultrasound, higher temperature only has a slight effect. Solid-to-liquid ratio does not have a positive or negative impact. For D18, the influence of ultrasound is measurable, but is not as strong as it is for the smaller D14. The difference between intermittent and permanent ultrasound irradiation seems negligible, but higher solid-to-liquid ratios have an adverse effect in the runs with intermittent ultrasound. With permanent ultrasound, this inhibition is apparently gone. Higher bath temperature, on the other hand, influenced leaching in a positive way. The secondary axis "expected leaching efficiency" has to be viewed with caution. The 100% mark refers to the expected 20% weight percent of leachable substance in the PCD. In fact, this value has only been orally confirmed by the manufacturer and the only analysis is the surface SEM and EDS, as shown at Figures 1 and 2. The leaching efficiency was calculated using Equation (1). The obtained results with D14 show that this value is reasonably accurate.

$$\text{Leaching efficiency [\%]} = (\Delta m_{PCD} \text{ or } c(Co)_{SOL})/(0.2(m_{0,PCD})) \tag{1}$$

where $\Delta m_{PCD}$ is the lost PCD weight after the experiment, $c(Co)_{SOL}$ is the cobalt content in solution after the experiment, 0.2 is the given factor of initial cobalt content in PCD and $m_{0,PCD}$ is the initial PCD weight.

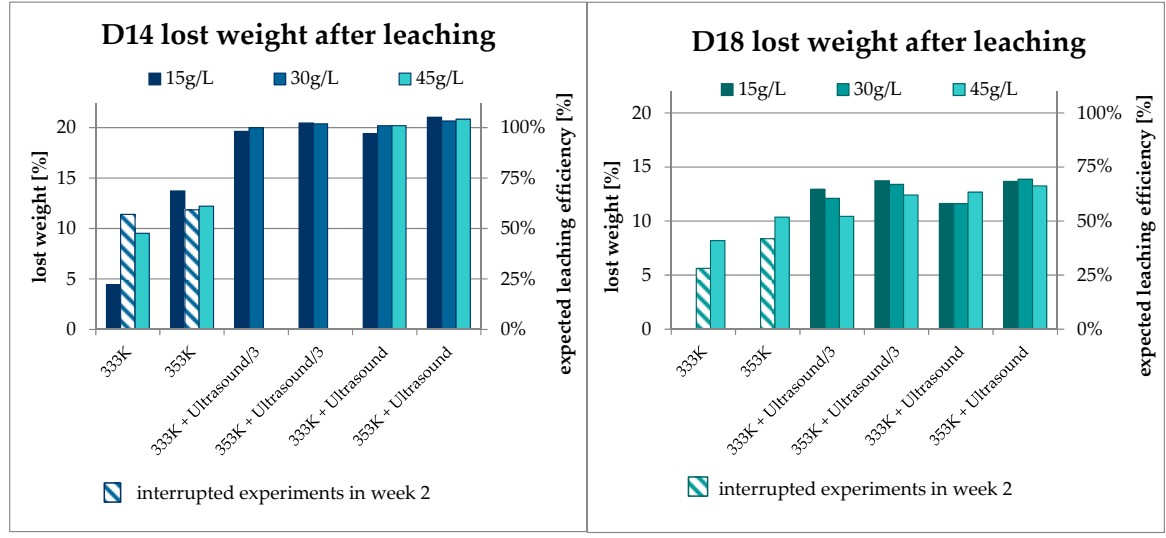

**Figure 10.** Lost weight for different PCD samples after leaching with different parameters described in the plot. 35 kHz ultrasound varied from "none", "ultrasound on" for a third of the run time, to "full-time ultrasound", as shown from left to right.

Weight data was used to calculate the average PCD weight lost per day, as shown at Figures 10 and 11. The difference between initial weights and final dry weights was divided by the actual batch time in hours. This was done to renormalize the values to a certain time interval because the batches had different run times. Though differences in the diagrams above may only be small, the diagrams below are truly adequate for a comparison. The negative effect of the solid-to-liquid ratio on leaching efficiency of cobalt from D18 blanks with intermittent ultrasound became clearer, as shown at Figure 11. A less obvious difference is the fact that the highest columns in the diagrams below now truly are the runs with the most extracted weight percentage per time.

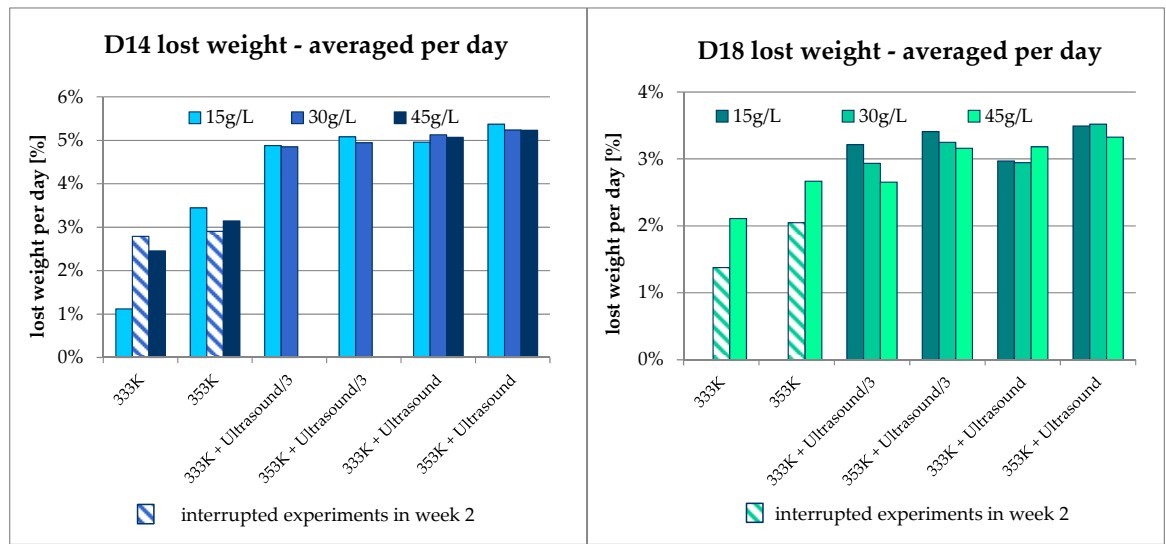

**Figure 11.** Lost weight for different PCD samples averaged per day. 35 kHz ultrasound varied from "none", "ultrasound on" for a third of the run time, to "full-time ultrasound", as shown from left to right.

In weeks 5, 8, and 9 the volume of remaining liquid was measured. The weight of dissolved cobalt was calculated by combining these volumes with the Co (II) concentrations from chemical analysis. A comparison between the weight of dissolved Co (II) versus lost PCD weight during leaching is presented in Figure 12. If pure metallic cobalt was used and remained in its metallic state during the

making of the PCD, there should be no difference between these values. All the material that is leached from the PCD should be cobalt and end up in the solution as dissolved ions.

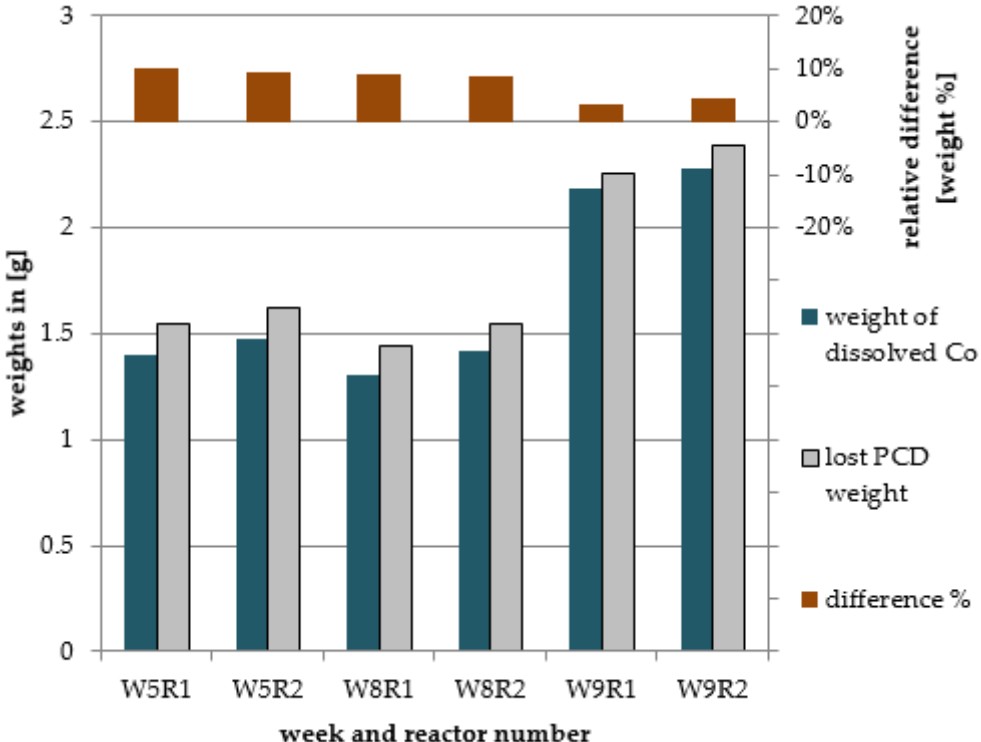

**Figure 12.** Weight of dissolved Co (II) versus lost PCD weight.

As can be clearly seen, there is a significant difference, ranging between +3% and +10%, among the compared values. On account of the EDS data (Table 1), this difference probably stems from cobalt oxides and iron impurities rather than fluctuations or measuring errors and uncertainties. Finally, the obtained solution using an aqua regia (Figure 13—left) from polycrystalline drawing die blanks via the ultrasound-assisted leaching process is shown in Figure 13—right. Cobalt powder was obtained from this solution using the precipitation method.

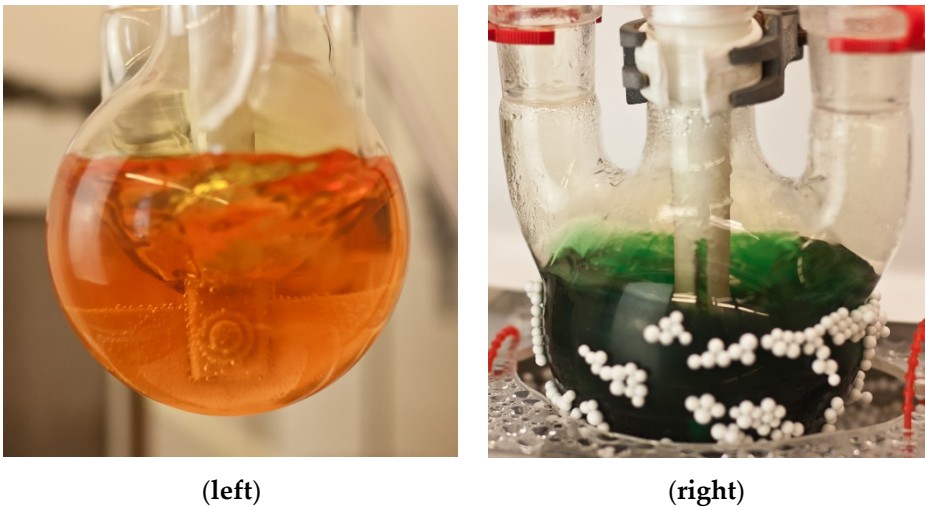

(**left**)                     (**right**)

**Figure 13.** (**Left**): Aqua regia approximately 9 min after mixing and stirring. (**Right**): View of the reactor during the experiment, containing the PCD and cobalt bearing solution. In this case, it was running at 333 K, with full-time ultrasound and 45 g/L solid-to-liquid ratio.

*3.2. Results Regarding Process Optimization*

The results from this study suggest that the leaching of D14 does not require a 353 K bath temperature but can be done at 333 K. Ultrasound can accelerate the leaching process to the extent that the PCD can reach a desaturated state with less than 10% of metallic inclusions remaining after three to four days if they are leached at low solid-to-liquid ratios close to 15 g/L. The depth of ultrasound penetration into PCD, meaning the depth at which the disruptive effects of ultrasound are no longer able to outrun diffusion, was determined to be in the region of 1.4–1.7mm.

If the emphasis were put on just shortening the leaching time, one way could be to leach at 353 K, replacing the solution after three days to reset the concentration gradient and refresh the active compounds in solution. As there is more research needed regarding the kinetics and mechanism of the dissolution process, the same applies to process safety and possible replacement of aqua regia with other less harmful leaching agents. The potentiometric aspects of leaching of cobalt from PCD also deserve considerable attention in order to ensure controlled potential cobalt leaching as a selective way for total cobalt removal in a short time. The kinetics and mechanisms of the studied ultrasound-assisted leaching process from polycrystalline diamond blanks will be reported in Part 2 [24] of this research in detail.

*3.3. Possible Recycling Routes for Cosolution in Order to Produce Cobalt Powder and Its Compounds*

After the experiments, a highly acidic aqua regia solution laden with divalent cobalt remains. This cobalt content is very valuable and should not be discarded. One simple (but hardly elegant) way to reuse the cobalt chloride would be the complete evaporation of liquids using the remaining cobalt chloride as drying salt. Instead, there are ways to selectively extract Co from the solution with DEHPA2, for example, and then precipitating or electrolytic winning of the metal powder. As stated earlier, there also is the possibility to make cobalt nanopowder using the ultrasonic spray pyrolysis method and the chemical reduction method in the aqueous solution, which would be a very versatile substance to be used in battery technology as well as catalyst applications. The production of cobalt hydroxide shall be reached using sodium hydroxide as precipitation agent. A goal-oriented refining process such as solvent extraction as a traditional hydrometallurgical method could be imagined depending on the desired metal powder and its compounds.

## 4. Conclusions

This study was designed for the recovery of pure demetallized PCD and cobalt from raw PCD. In nine experimental runs with a 5 day duration, cobalt containing PCD was leached in aqua regia at atmospheric pressure between 333 K and 353 K. Using two reactors in parallel, temperature, ultrasound irradiation time, solid-to-liquid ratio, and PCD size were varied to find out which parameters are beneficial and could possibly accelerate this process. PCD weights and cobalt content in solution were also monitored. It was found that aqua regia accumulated more dissolved cobalt at 333 K than at 353 K probably due to volatile reagents being less available over time. The ultrasound treatment increases the leaching efficiency. With added ultrasound (even at just a third of total run time) and at a low S/L ratios close to 15 g/L, the leaching time for D14 to reach the 90% leached mark was reduced to three days, which is a significant shortening of leaching time. PCD type D18 with a thickness of 3.5 mm was not leached to completion within five days. The leaching temperature had more impact on the results than ultrasound. These findings were reinforced by the mass balance in which a small discrepancy was found. The PCD lost a fraction of weight that could not be explained by the weight of dissolved cobalt. From EDS data and the nature of PCD, this fraction probably consisted of oxygen from oxides in the PCD or single diamond grains that were broken off by the impact of ultrasound. Advances in synthesis of metallic powders using the ultrasound-assisted leaching process from polycrystalline diamond blanks can be used for the cemented tungsten carbide in order to estimate a scale-up of this process in future.

**Recovery of Diamond and Cobalt Powders from Polycrystalline Drawing Die Scraps via Ultrasound-Assisted Leaching Process—Part 2: Kinetics and Mechanisms**

The kinetic models were used for the study of cobalt dissolution from polycrystalline diamond blanks via a measurement of declining ferromagnetic properties over time. For a better understanding of this leaching process, thermochemical aspects were included in this work. The lowest free Gibbs energy corresponds to a low solid/liquid ratio and fully used ultrasound in the process. A transition from a reaction-controlled to a diffusion-controlled shrinking core model was found for PCD with a thickness larger than 2.8–3.4 mm. Intermittent ultrasound doubles the reaction rate constant and fully using of ultrasound causes a further increase with a factor of 1.5. The obtained activation energy between 333 K and 353 K is 20 kJ/mol, and small for all diamond blanks with a diameter size of 5 μm, which corresponds to the diffusion-controlled process.

**Author Contributions:** F.K. and S.S. conceptualized and managed the research. S.S. cowrote the paper. S.G. contributed the SEM and EDS analysis of the PCD surface. B.F. supervised personnel, coordinated resources, and co-wrote the paper. F.K. performed the experiments and wrote the paper. All authors have read and agreed to the published version of the manuscript.

**Funding:** This research was funded by Projektträger Jülich (PtJ), Grant Number 005-1902-0147.

**Acknowledgments:** We would like to thank Redies Deutschland GmbH & Co. KG (Aachen, Germany) for providing PCD samples as well as additional equipment.

**Conflicts of Interest:** The authors declare no conflict of interest.

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
