# Peer review of "Recovery of Diamond and Cobalt Powder from Polycrystalline Drawing Die Blanks via Ultrasound-Assisted Leaching Process—Part 1: Process Design and Efficiencies"

_metals, doi:10.3390/met10060731_

Round 1

Reviewer 1 Report

Dear autors,

this paper is interesting since it seems new idea to recover diamond, cobalt and so on. this new topic can be published after following minor corrections.

Abstract: "The" is bold style.

Figure1, 2, scale is difficult to see.

Figure 14, it is difficult to see. it should be more focused photo.

Author Response

Dear reviewer, thank you very much for your valuable comments and invested time in order to improve our first Version. Our answers are included in new improved Version and cover letter..

"Dear authors,

this paper is interesting since it seems new idea to recover diamond, cobalt and so on. this new topic can be published after following minor corrections".

1) Abstract: "The" is bold style.

I changed the bold style for „The“ in Abstract

2) Figure1, 2, scale is difficult to see.

Now is the scale of 1 µm is visible on the Figure 1 and Figure 2.

“In Figure 2 the dark grey areas are the bridged diamond grains with cavities where also traces of cobalt show up in lighter shades.” was added.

3) Figure 14, it is difficult to see. it should be more focused photo.

The figure 14 is removed from our text. It is not possible to find one excellent picture for this material after leaching process.

Reviewer 2 Report

The paper is an interesting contribution and the work described is well done. I would like to comment on the writing style, however. The paper could be more direct, it was not clear at first that cobalt was being leached out of synthetic polycrystalline diamond. The initial discussion of cobalt ores was a misleading distraction that did not add much to the value of the paper. I would recommend commencing with a direct discussion of why cobalt is in the PCD, and why it needs to be removed, without a digression about mining and laterite deposits. 

On page 5, line 136, there is a broken reference link.

In general, I recommend that the authors consider making the rest of the paper more direct. The main issue is whether or not ultrasonic treatment increases the leaching rate, and so the value of the paper could be improved further by making sure that all of the data presented relates to that question.

Author Response

Dear Reviewer, thank you very much for your invested time ad your valuable comments in order to improve this paper. You can find our answers in new improved text and our cover letter.

1) "The paper is an interesting contribution and the work described is well done. I would like to comment on the writing style, however. The paper could be more direct, it was not clear at first that cobalt was being leached out of synthetic polycrystalline diamond. The initial discussion of cobalt ores was a misleading distraction that did not add much to the value of the paper. I would recommend commencing with a direct discussion of why cobalt is in the PCD, and why it needs to be removed, without a digression about mining and laterite deposits."

We changed it: 

Cobalt is used as a solvent catalyst in the production of PCD that would otherwise take even more pressure and a higher temperature to achieve. [4]. The wide use of cobalt relative to other metals is associated with the high solubility of carbon in its melt during thermobaric treatment [5]. Unfortunately, at temperatures above 800°C, which are developed during operation of a polycrystalline-diamond tool, the difference in thermal expansion coefficients of cobalt and diamond promotes the formation of microcracks and results in significant heat-resistance reduction and therefore reduction of the abrasion resistance of the polycrystalline diamond [6]. Thus, the reduction of the cobalt content in the sintered polycrystalline diamond (PCD) greatly improves the performance of a superhard composite. Finally, removal of cobalt from the used PCD is most important request in industry of wire drawing tools made from diamond materials and tungsten carbide.

We put new references:

  1. Anokhin, A.S., Strel’nikova, S. S, Andrianov, M.A., Tkachenko, V.V., Shipkov, A.N., Kukueva,, E.V., Gol’dt, A.E., O. T. Zaremba,, Formation of the Structure and Phase Composition of Polycrystalline Diamond with Cobalt Infiltration in the System Diamond – Hard Alloy, Glass and Ceramics 2019, 75, 475–478
  2. Wentorf, R.H., DeVries, R.C, Bundy, F.P. Sintered superhard materials. Science 1980, 208 (4446), 873 – 880
  3. Scott, T.A, The influence of microstructure on the mechanical properties of polycrystalline diamond: a literature review, Adv. Appl. Ceram. 2018, 117 (3), 161 – 176.

Cobalt is used as a solvent catalyst in the production of PCD that would otherwise take even more pressure and a higher temperature to achieve.

2) On page 5, line 136, there is a broken reference link.

We removed it

3) In general, I recommend that the authors consider making the rest of the paper more direct. The main issue is whether or not ultrasonic treatment increases the leaching rate, and so the value of the paper could be improved further by making sure that all of the data presented relates to that question.

The ultrasound treatment increases the leaching efficiency

With added ultrasound and at low S/L ratios close to 15g/L, the leaching time for D14 to reach the amount of 90% leached mark was reduced to three days, a significant shortening of leaching time; even when using ultrasound just for a third of the total run time. 

Reviewer 3 Report

I think this article has a very important  topic for world science. The text was prepared  not very carefully. In the paper was presesnted the treatment of industrial polycrystalline diamond (PCD) blanks in aqua regia at atmospheric pressure between 60-80°C was performed via ultrasound assisted leaching to investigate.
I have a few questions of the presented process.
-  most of figures presented in the text  are not clear  presented, for example:
figure 4 - what is presented in picture and why?
figure 5 - very weak quality  figure
figure 6  - what is presented in picture and why? I think is not important picture for this article - I propose delete
figure 7 - very weak quality  figure
figure 8 - in all text the Authors used two type of temperature units - I propose used Kelvin's units but in all text, fugures and tables uniformly
figure 9 - the same in figure 8 and a question  about what is full ultrasound - I ask about parameters?
figure 10  - the same in figures 8 and 9 about temperature units
figure 11 - the presented the results are not clearly - must be improved
figure 12  - better nor figure 11 but what are ultrasound and ultrasound /3 - must be showed parameters in the figure
figure 13 - must be improved - unreadable
figure 14 - very weak quality  figure
I have a question about  what the second part of the article will be about?
The final question concerns for the economically aspects the process?
Summary:
Overall, the research method and analysis of results are well-documented and make sense. The authors presented modest introduction literature overview, too. The subject and the methodology applied seem to be justified and deserve publication but after major reviewers. The topic is very important in world.

Author Response

Dear Reviewer, thank you very much for your invested time ad your valuable comments in order to improve this paper. You can find our answers in new improved text and our cover letter.

-I think this article has a very important topic for world science. The text was prepared not very carefully. In the paper was presented the treatment of industrial polycrystalline diamond (PCD) blanks in aqua regia at atmospheric pressure between 60-80°C was performed via ultrasound assisted leaching to investigate.

I have a few questions of the presented process.
-  most of figures presented in the text  are not clear  presented, for example:

1) figure 4 - what is presented in picture and why?

At figure 4 we presented the leaching reactor and most important parts. The real picture of the reactor is replaced with one with better resolution. The leaching reactor is most important part of our leaching strategy. Therefore we offered it with one description.

 2) figure 5 - very weak quality  figure

We put new Figure with better resolution.

 3) figure 6  - what is presented in picture and why? I think is not important picture for this article - I propose delete

We removed it.

4) figure 7 - very weak quality  figure

We offered the better resolution in our new Version

 5) figure 8 - in all text the Authors used two type of temperature units - I propose used Kelvin's units but in all text, figures and tables uniformly

We accepted it. We will use Kelvin's units in all text, figures and tables uniformly

 6) figure 9 - the same in figure 8 and a question  about what is full ultrasound - I ask about parameters?

The ultrasound bath has 60W average ultrasound power. The output level in these Bandelin devices is fixed. So the ultrasound irradiation was altered by a) not using ultrasound b) using ultrasound only a third of the experimental time c) by leaving ultrasound running all the time.

“Bandelind Sonorex RK 52 H” … “These ultrasound baths have a nominal frequency of 35kHz and put out 60W effectively while output peaks can happen up to 240W. The output level is fixed and ultrasound irradiation was altered by using it intermittently.”

Caption for Fig.9 changed: “Concentration of cobalt in solution in time between 333 K and 353 K with full-time ultrasound.”

 7) figure 10  - the same in figures 8 and 9 about temperature units

Concentration of cobalt in solution in time between 333 K and 353 K in the fully presence of ultrasound.

 8) figure 11 - the presented the results are not clearly - must be improved

We changed of size of text and convert °C in K

 9) figure 12  - better nor figure 11 but what are ultrasound and ultrasound /3 - must be showed parameters in the figure

Shorthand for “using ultrasound only a third of the experimental run time”

10) figure 13 - must be improved - unreadable
Figure 13 is improved.

11) figure 14 - very weak quality  figure
Figure 14 is removed from text.

12) I have a question about, what the second part of the article will be about?

Recovery of diamond and cobalt powders from polycrystalline drawing die scraps via ultrasound assisted leaching process – Part 2: Kinetics and mechanisms

The obtained activation energy between 333 K and 353 K is 20 kJ/mol and below this value, what corresponds a diffusion controlled process.

13) The final question concerns for the economically aspects the process?

Cobalt belongs to the critical metals. REDIES GmbH, Aachen produces wire drawing tools made from diamond materials and tungsten carbide, and needs recycling technology for the dissolution of cobalt. After this treatment, PCD are returned in production process.

The customers using Redies dies raised demand for thermally stable drawing tools. As an example, Plansee in Liezen hot-draws Niobium and Tantalum wire. Regular PCD would break almost instantly and not be cost efficient nor environmentally friendly. Leaching with aqua regia to get thermostable PCD is economically sound, yet very unfriendly for the environment, so the focus for future research should be on finding a better leaching solution and maybe producing cheaper PCD, possibly without using cobalt but iron or nickel instead. It is possible to use iron or nickel for the production of PCD, yet the quality is still inferior. As of now, CVD and other methods are still not able to compete with the high-temperature-high-pressure sintering process.

Reviewer 4 Report

This Manuscript reports a research aimed at extraction of metal, particularly cobalt, values from polycrystalline diamond blanks using aqueous regia leaching assisted by ultrasonic treatment. The results are new and interesting, and fall into the Journal scope. In general, the paper is rather well written, although somewhat voluminous, and its organization is unusual as 10 of 14 figures belong to Experimental section. Some of these details look excessive and confusing; for example, I didn’t understand how the magnetic measurements were applied in the leaching study. I also have reservations regarding the discussion in paragraphs 3.2 and especially 3.3 about recycling the Co-bearing solutions and production of “cobalt nanopowder”. This is fully speculative, as even the composition of aqua-regia-based media is not specified.  I would recommend shortening the above, but this can be left at the authors’ responsibility. It is also unclear what means “Recovery of diamond..” in the title.

The paper can be published after addressing several points.

1) Something is wrong with Figure 5 proportions, resolution and caption, please correct.

2) Almost all figure captions are too short, and should be made more informative, allowing to understand the matter without exploring the text. Please also check the line 260

3) The titles and units at abscissa axes are sometimes uncommon and confusing (“in %”, “in [g]”, etc.), and different for similar plots. Please specify, whether “difference [%]” in Fig. 13 are relative or absolute wt%.

4) Figure 12 shows “Lost weight for different PCD -sample averaged per day”. Meanwhile, the kinetic curves in Figs. 8-10 are notably non-linear. How were the lost weight values calculated?

5) Please explain what is depicted in Fig. 14 (unfortunately, very indistinct), in particular, what is the length scale.  

Author Response

Dear Reviewer, thank you very much for your valuable comments and invested time in order to omprove this paper. We have included our answers in our improved Version and cover letter.

"This Manuscript reports a research aimed at extraction of metal, particularly cobalt, values from polycrystalline diamond blanks using aqueous regia leaching assisted by ultrasonic treatment. The results are new and interesting, and fall into the Journal scope. In general, the paper is rather well written, although somewhat voluminous, and its organization is unusual as 10 of 14 figures belong to Experimental section. Some of these details look excessive and confusing; for example,

  1. I didn’t understand how the magnetic measurements were applied in the leaching study.

Magnetic measurements are performed in order to follow the magnetic characteristics of PCD during the removal of cobalt in time. The obtained values are compared with initial values and the recovery of cobalt was compared via leaching efficiency.

There was no way to know the Cobalt content in the PCD without destroying them for samples. Except, Cobalt is ferromagnetic and can influence a magnetometer. That way the progress of leaching was monitored in the experiments. The goal in the industrial process is complete demetallization while keeping the tool material intact.

  1. I also have reservations regarding the discussion in paragraphs 3.2 and especially 3.3 about recycling the Co-bearing solutions and production of “cobalt nanopowder”.

This is fully speculative, as even the composition of aqua-regia-based media is not specified.  I would recommend shortening the above, but this can be left at the authors’ responsibility. It is also unclear what means.

We have just prepared nanosized cobalt particles from waste WC-Co powders using ultrasonic spray pyrolysis, as shown in our paper in Materials Research Bulletin. Adjusting the concentration of cobalt chloride solution, it is possible an ultrasonic spray pyrolysis with hydrogen reduction at 900°C to obtain cobalt nanosized powder. We have just proposed it to company Redies GmbH Aachen, which have this waste material is big problem.

  • Gürmen, S., Stopić, S, Friedrich, B. Synthesis of nanosized spherical cobalt powder by ultrasonic spray pyrolysis method, Materials Research Bulletin 2006, 41, 10, 1882-1890.
  • “More specifically, the aqua regia was mixed from 3 parts Merck KGaA fuming hydrochloric acid 37%, Emsure ACS/ISO quality and one part PanReac ApplicChem nitric acid 65% ISO analysis quality.” was added in line 53.

3) “Recovery of diamond..” in the title.

Recovery of diamond means removal of cobalt from diamond and return of PCD in production process. Title was changed.

4) The paper can be published after addressing several points.

  1. Something is wrong with Figure 5 proportions, resolution and caption, please correct.

We correct this resolution (as shown in our letter to reviewer and editors)

5) Almost all figure captions are too short, and should be made more informative, allowing to understand the matter without exploring the text. Please also check the line 260

Line 260 was corrected “These measurements were used to compare the results of all runs, shown individually for D14 and D18 blanks in Figure 11. At first glance, the most obvious fact is that the larger PCD were not leached to completion within the 90 to 100 hours timeframe.”

6) The titles and units at abscissa axes are sometimes uncommon and confusing (“in %”, “in [g]”, etc.), and different for similar plots. Please specify, whether “difference [%]” in Fig. 13 are relative or absolute wt%.

Data plot and axis title corrected.

7) Figure 12 shows “Lost weight for different PCD -sample averaged per day”. Meanwhile, the kinetic curves in Figs. 8-10 are notably non-linear. How were the lost weight values calculated?

Initial weight minus final weight of the (dried) PCD. This difference is then divided by the number of hours over which the experiment was live, giving an average weight loss per day. (Text was also edited.)

8) Please explain what is depicted in Fig. 14 (unfortunately, very indistinct), in particular, what is the length scale.  

Image and caption changed. This is now Figure 13.“Left: Aqua regia approximately 9 minutes after mixing and stirring. Right: view of reactor during experiment, containing PCD and cobalt bearing solution. In this case running at 333K, with full ultrasound and 45g/l solid-to-liquid ratio.”

Round 2

Reviewer 3 Report

Dear Autors, thank you very much for your answer and improvement this paper. I confirm this article has a very important topic for world science. The text was improvemented in few steps. In the paper was presented the treatment of industrial polycrystalline diamond (PCD) blanks in aqua regia at atmospheric pressure between 60-80°C was performed via ultrasound assisted leaching to investigate.